# RIEMANNIAN MANIFOLD EMBEDDINGS FOR STRAIGHT-THROUGH ESTIMATOR

## ABSTRACT

Quantized Neural Networks (QNNs) aim at replacing full-precision weights $\boldsymbol{W}$ with quantized weights $\hat{\boldsymbol{W}}$, which make it possible to deploy large models to mobile and miniaturized devices easily. However, either infinite or zero gradients caused by non-differentiable quantization significantly affect the training of quantized models. In order to address this problem, most training-based quantization methods use Straight-Through Estimator (STE) to approximate gradients $\nabla_{\boldsymbol{W}}$ w.r.t. $\boldsymbol{W}$ with gradients $\nabla_{\hat{\boldsymbol{W}}}$ w.r.t. $\hat{\boldsymbol{W}}$ where the premise is that $\boldsymbol{W}$ must be clipped to $[-1, +1]$. However, the simple application of STE brings with the gradient mismatch problem, which affects the stability of the training process. In this paper, we propose to revise an approximated gradient for penetrating the quantization function with manifold learning. Specifically, by viewing the parameter space as a metric tensor in the Riemannian manifold, we introduce the Manifold Quantization (ManiQuant) via revised STE to alleviate the gradient mismatch problem. The ablation studies and experimental results demonstrate that our proposed method has a better and more stable performance with various deep neural networks on CIFAR10/100 and ImageNet datasets.

## 1 INTRODUCTION

Neural networks can handle many complex tasks due to their large number of trainable parameters and strong nonlinear capabilities (Krizhevsky et al., 2012). However, the massive amount of models and calculations hinder the application of neural networks on mobile and miniaturized devices, which naturally comes with constraints on computing power and resources. Neural network quantization is considered an efficient solution in the inference that alleviates the number of parameters and optimizes the computation by reducing the bit width of weights or activations (Courbariaux et al., 2016; Li et al., 2016; Zhu et al., 2016).

Existing neural network quantization methods can be roughly divided into two categories: "STE" and "Non-STE" methods. Most of the quantization methods adopted by the QNNs belong to the former, i.e. there is always a non-differentiable quantization function during training. The role of STE is to penetrate this non-differentiable quantization function and pass the gradients in backpropogation (Hinton, 2012), e.g. DeepShift (Elhoushi et al., 2019), INT8 (Zhu et al., 2020), AQE (Chen et al., 2020), etc. "Non-STE" methods refer to maintain feasible quantization during training, which does not need to apply STE directly to all full-precision weights. For example, Zhou et al. (Zhou et al., 2017) divided the weights into two groups until all parameters are quantized, where the first group is directly quantized and fixed; the second group needs to be retrained to make up for the decrease of accuracy caused by quantization of the first group. Louizos et al. (Louizos et al., 2019) introduced a differentiable quantizer that can transform the continuous distributions of weights and activations to categorical distributions with gradient-based optimization. However, "Non-STE" methods face the setting and influence of heavy hyper-parameters in the training process.

Relatively, "STE" methods are wider choices for quantized models from simplicity and versatility. To approximate $\nabla_{\boldsymbol{W}}$ w.r.t. $\boldsymbol{W}$ and complete stable quantization training, Courbariaux et al. (Courbariaux et al., 2016) binarized the neural networks by the approximated gradients using STE:

$$\nabla_{\boldsymbol{W}} = \nabla_{\hat{\boldsymbol{W}}} \circ \mathbb{I}, \quad \text{where } \frac{\partial \hat{\boldsymbol{W}}}{\partial \boldsymbol{W}} = \mathbb{I} = \left\{ \begin{array}{ll} 1 & \text{if } |\boldsymbol{W}| \le 1 \\ 0 & \text{otherwise} \end{array} \right. .$$

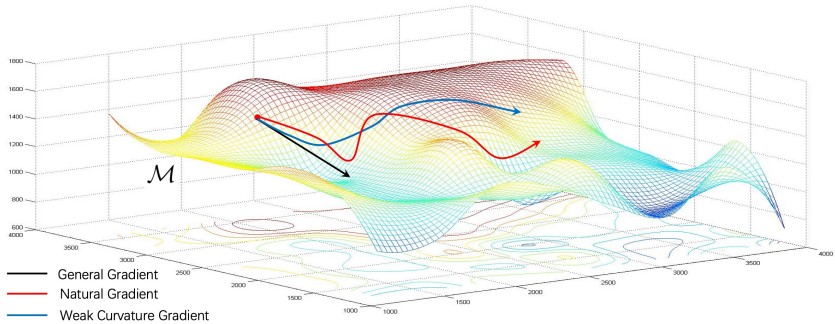

Figure 1: The general gradient is intuitively defined in Euclidean space, which means that the direction of the general gradient will not be affected by the curvature. On the contrary, the natural gradient and weak curvature gradient intrinsically contain the curvature of manifolds, although the definitions of the manifolds constructing these two gradients are different.

However, it will inevitably bring the gradient mismatch problem raised by just simply application of STE. To overcome this challenge, Zhou et al. (Zhou et al., 2016) proposed to transform $\boldsymbol{W}$ to $\tilde{W}$:

$$\tilde{\boldsymbol{W}} = \frac{\tanh(\boldsymbol{W})}{\max(|\tanh(\boldsymbol{W})|)},$$

and then quantize it using a quantization function $Q(\cdot)$. During backpropogation, the gradients $\nabla_{\boldsymbol{W}}$ w.r.t. $\boldsymbol{W}$ can be further computed using the chain rule:

$$\nabla_{\boldsymbol{W}} = \nabla_{Q(\tilde{\boldsymbol{w}})} \frac{1 - \tanh^2(\boldsymbol{W})}{\max(|\tanh(\boldsymbol{W})|)}.$$

Based on this work, Chen et al. (Chen et al., 2019) proposed to learn $\nabla_{\boldsymbol{W}}$ by fully-connected neural networks or LSTM, which replaces this gradients via a meta quantizer $M_\phi$ parameterized by $\phi$ across layers:

$$\nabla_{\boldsymbol{W}} = M_\phi\left(\nabla_{Q(\tilde{\boldsymbol{W}})}, Q(\tilde{\boldsymbol{W}})\right) \frac{\partial \tilde{\boldsymbol{W}}}{\partial \boldsymbol{W}}.$$

However, such methods not only add many additional parameters, but also increase the difficulty of the training of quantized models.

In this paper, we introduce the Manifold Quantization (ManiQuant) to train a quantized model via embedding Riemannian manifolds for STE. In Figure 1, we treat the parameter space in a quantized model as a Riemannian manifold, which will alleviate the gradient mismatch problem caused by non-differentiable quantization and assist quantized models in achieving a more stable convergence and better performance. The main contributions of this work are three-fold: **First**, we propose to use Fisher Information Matrix embedding to alleviate the gradient mismatch problem. **Second**, we define a novel Hyperbolic divergence by a convex function with geometric structure. With the constraint of Hyperbolic divergence, we introduce a weak curvature manifold that forms the background of ManiQuant. **Third**, based on the **Second**, we propose to use weak curvature metric embeddings for STE as the weak curvature gradient to approximate $\nabla_{\boldsymbol{W}}$ without extra parameters and training complexity.

## 2 RELATED WORK

### 2.1 GENERAL GRADIENT

For a neural network with $l$ layers, the full-precision weight matrix of each layer is just marked as $\boldsymbol{W}_i$. By defining the operation $\text{vec}(\cdot)$ that vectorizes matrices by stacking their columns together, the total parameter vector of the neural network can be denoted as $\boldsymbol{\theta} = \left[\text{vec}\left(\boldsymbol{W}_1\right)^\top, \text{vec}\left(\boldsymbol{W}_2\right)^\top, \ldots, \text{vec}\left(\boldsymbol{W}_l\right)^\top\right]^\top$. The parameter vector is given as a column vector.

Let $\boldsymbol{\theta} \in \mathcal{R}^n$ be a parameter space on which a loss function $L$ associated with weight is well-defined. It is relatively easy to express the general gradient in training:

$$\nabla_{\boldsymbol{\theta}} L = \frac{\partial L}{\partial \boldsymbol{\theta}}, \tag{1}$$

which is the steepest descent method in Euclidean space with an orthonormal coordinate. Note that the negative gradient represents the direction of the steepest descent.

When the Euclidean space is considered, the Euclidean divergence between two sufficiently closed points $\boldsymbol{\theta}$ and $\boldsymbol{\theta}'$ is actually defined by default:

$$D_E[\boldsymbol{\theta} : \boldsymbol{\theta}'] = \frac{1}{2} \sum_i (\theta_i - \theta'_i)^2, \tag{2}$$

which is a half of the square of the Euclidean distance identified by the Euclidean metric $\delta_{ij}$, so that

$$ds^2 = 2D_E[\boldsymbol{\theta} : \boldsymbol{\theta} + d\boldsymbol{\theta}] = \sum_i (d\theta_i)^2 = \sum_{i,j} \delta_{ij} d\theta_i d\theta_j. \tag{3}$$

## 2.2 Natural Gradient

The general gradient only considers the parameter update along the gradient direction that does not involve the metric tensor for the parameter space of the problem. Now, we attach a Riemannian metric $G_{ij}(\boldsymbol{\theta})$ for the parameter space to form a Riemannian manifold $(\mathcal{M}, G_{ij})$ (Amari, 1998; 2016). In that case, the steepest descent direction also depends on the quadratic form introduced by a small incremental vector $d\boldsymbol{\theta}$ that connects $\boldsymbol{\theta}$ and $\boldsymbol{\theta} + d\boldsymbol{\theta}$, whose form is given by

$$ds^2 = \sum_{i,j} G_{ij}(\boldsymbol{\theta}) d\theta_i d\theta_j, \tag{4}$$

where $d\theta_i$ is the component of $d\boldsymbol{\theta}$. Under the constraint $ds^2$, the steepest descent direction is toward the optimization goal of $L(\boldsymbol{\theta} + d\boldsymbol{\theta})$. Intuitively, it measures the Kullback-Leibler (KL) divergence between two distributions $p(\boldsymbol{x}|\boldsymbol{\theta})$ and $p(\boldsymbol{x}|\boldsymbol{\theta} + d\boldsymbol{\theta})$ of network model, which is equivalent to the interval of two adjacent points on a Riemannian manifold. The KL divergence under the Riemannian metric is well approximated through the Fisher Information Matrix (FIM) with the second-order Taylor expansion of the KL divergence (see Appendix B.3) (Ba et al., 2016):

$$ds^2 = 2D_{KL}[p(\boldsymbol{x}|\boldsymbol{\theta}) : p(\boldsymbol{x}|\boldsymbol{\theta} + d\boldsymbol{\theta})] \approx -d\boldsymbol{\theta}^\top \mathbb{E}_{p(\boldsymbol{x}|\boldsymbol{\theta})}[\nabla \log p(\boldsymbol{x}|\boldsymbol{\theta}) \nabla \log p(\boldsymbol{x}|\boldsymbol{\theta})^\top] d\boldsymbol{\theta}. \tag{5}$$

We see that FIM is equal to the negative expected Hessian of log likelihood. Furthermore, Amari deduced that the Riemannian metric is given by the FIM (Amari, 1998). By the Lagrangian form, we have the natural gradient (see Appendix C.1)

$$\tilde{\nabla}_{\boldsymbol{\theta}} L = F^{-1}(\boldsymbol{\theta}) \frac{\partial L}{\partial \boldsymbol{\theta}}, \quad \text{where } F(\boldsymbol{\theta}) = \mathbb{E}_{p(\boldsymbol{x}|\boldsymbol{\theta})}[\nabla \log p(\boldsymbol{x}|\boldsymbol{\theta}) \nabla \log p(\boldsymbol{x}|\boldsymbol{\theta})^\top], \tag{6}$$

which is the steepest descent method in a Riemannian space and $F(\boldsymbol{\theta})$ is the FIM with a parameter vector (the natural gradient also known as the Riemannian gradient). Empirically, the immediate application of FIM is as drop-in replacement of Hessian in second order optimization algorithm. Note that $\tilde{\nabla}_{\boldsymbol{\theta}} L$ will return to $\nabla_{\boldsymbol{\theta}} L$ when $G_{ij}(\boldsymbol{\theta})$ is equal to the Euclidean metric $\delta_{ij}$, i.e. the identity matrix $\boldsymbol{I}$.

## 3 Manifold Quantization

Considering a quantized model of each layer, we notate the quantized weight matrix using $\hat{\boldsymbol{W}}_i$ for differentiating full-precision weight matrix $\boldsymbol{W}_i$. Note that we define the quantized parameter vector as $\hat{\boldsymbol{\theta}}$, which is similar to the full-precision parameter vector $\boldsymbol{\theta}$. During training a QNN, the quantization function $Q(\cdot)$ is a one-to-one mapping from full-precision values to quantized values, which can be expressed as $\hat{\boldsymbol{\theta}} = Q(\boldsymbol{\theta})$.

By involving the practice to train a neural network with low bit-width, the process of the quantization needs to be further designed that plays a vital role in the final performance, as for the other parts, e.g. via mapping from an input pattern $\hat{a}_{i-1}$ to output $\hat{a}_i$ can still be imitated in the same way as full-precision neural networks:

$$
\begin{aligned}
\boldsymbol{s}_i &= \hat{\boldsymbol{W}}_i \hat{\boldsymbol{a}}_{i-1}, \\
\hat{\boldsymbol{a}}_i &= Q(f_i \circ \boldsymbol{s}_i),
\end{aligned}
\tag{7}
$$

where $f_i$ is non-linear function acted on element-wise. To distinguish the full-precision model, we mark the notation " ˆ " to represent the operation through a quantization function in a quantized model, whether for weight matrix $\hat{\boldsymbol{W}}_i$ or activation vector $\hat{\boldsymbol{a}}_i$.

## 3.1 FISHER INFORMATION MATRIX EMBEDDING FOR STE

The concept of the natural gradient is closely related to the FIM and KL divergence (see Appendix B.3). Since the KL divergence is intrinsic, the natural gradient is also intrinsically invariant under the parameter transformation. By viewing the FIM as an $l$ by $l$ block matrix where $l$ denotes the number of layers in a neural network, the natural gradient formula can be introduced to alleviate the gradient mismatch problem when training a QNN and updating its parameters.

**Lemma 1** *In the Riemannian manifold defined by KL divergence, Fisher Information Matrix embedding for Straight-Through Estimator is written by:*

$$
\begin{aligned}
\tilde{\nabla}_{\boldsymbol{\theta}} L &\overset{\text{STE}}{=} F^{-1}(\hat{\boldsymbol{\theta}}) \nabla_{\hat{\boldsymbol{\theta}}} L \circ \mathbb{I}_{|\boldsymbol{\theta}| \leq 1} \\
&= \mathbb{E}^{-1} \left[ \frac{d \log p(\boldsymbol{x}|\hat{\boldsymbol{\theta}})}{d\hat{\boldsymbol{\theta}}} \frac{d \log p(\boldsymbol{x}|\hat{\boldsymbol{\theta}})}{d\hat{\boldsymbol{\theta}}}^{\top} \right] \nabla_{\hat{\boldsymbol{\theta}}} L \circ \mathbb{I}_{|\boldsymbol{\theta}| \leq 1} \\
&= \mathbb{E}^{-1} \left[ \text{vec} \left( \frac{\partial L}{\partial \hat{\boldsymbol{\theta}}} \right) \text{vec} \left( \frac{\partial L}{\partial \hat{\boldsymbol{\theta}}} \right)^{\top} \right] \nabla_{\hat{\boldsymbol{\theta}}} L \circ \mathbb{I}_{|\boldsymbol{\theta}| \leq 1}.
\end{aligned}
\tag{8}
$$

*Furthermore, $F^{-1}(\hat{\boldsymbol{\theta}})$ can be expressed as*

$$
\begin{bmatrix}
\mathbb{E} \left[ \text{vec} \left( \frac{\partial L}{\partial \hat{\boldsymbol{W}}_1} \right) \text{vec} \left( \frac{\partial L}{\partial \hat{\boldsymbol{W}}_1} \right)^{\top} \right] & \cdots & \mathbb{E} \left[ \text{vec} \left( \frac{\partial L}{\partial \hat{\boldsymbol{W}}_1} \right) \text{vec} \left( \frac{\partial L}{\partial \hat{\boldsymbol{W}}_l} \right)^{\top} \right] \\
\vdots & \ddots & \vdots \\
\mathbb{E} \left[ \text{vec} \left( \frac{\partial L}{\partial \hat{\boldsymbol{W}}_l} \right) \text{vec} \left( \frac{\partial L}{\partial \hat{\boldsymbol{W}}_1} \right)^{\top} \right] & \cdots & \mathbb{E} \left[ \text{vec} \left( \frac{\partial L}{\partial \hat{\boldsymbol{W}}_l} \right) \text{vec} \left( \frac{\partial L}{\partial \hat{\boldsymbol{W}}_l} \right)^{\top} \right]
\end{bmatrix}^{-1}
\tag{9}
$$

*where* $\text{vec} \left( \frac{\partial L}{\partial \hat{\boldsymbol{W}}_i} \right)$ *can be represented by the gradient error $\frac{\partial L}{\partial \boldsymbol{a}_i}$.*

*Proof.* The proofs are the combination between STE and Appendix C.1. □

Considering that the gradient propagation needs to span over the quantized neurons and graphs, we still need to use STE to update the gradient of QNNs in the learning procedure:

$$
\text{vec} \left( \frac{\partial L}{\partial \hat{\boldsymbol{W}}_i} \right) = \hat{\boldsymbol{a}}_{i-1}^{\top} \otimes \left( \frac{\partial L}{\partial \boldsymbol{a}_i} \circ f_i'(\boldsymbol{s}_i) \right) \overset{\text{STE}}{=} \hat{\boldsymbol{a}}_{i-1}^{\top} \otimes \frac{\partial L}{\partial \hat{\boldsymbol{a}}_i} \circ \left( f_i'(\boldsymbol{s}_i) \mathbb{I}_{|\boldsymbol{a}_i| \leq 1} \right),
\tag{10}
$$

where $\otimes$ denotes the Kronecker product [1]. Relying on the KL divergence, we indicate the parameter space as the Riemannian manifold rather than the Euclidean space in the quantization procedure.

For neural networks with a scale of one million or more parameters, the time complexity of inverting the FIM, a component of natural gradients, is $O(n^3)$ (Povey et al., 2014). Previously, there were some works to calculate the natural gradient efficiently, e.g. Roux et al. (Roux et al., 2008) decomposed the FIM into multiple diagonal blocks, where each diagonal block is approximated by a low-rank matrix. Bastian et al. (Bastian et al., 2011) also used the idea of diagonal blocks by constructing a diagonal block that corresponds to a weight matrix. Martens et al. (Martens & Grosse,

---

[1]Since the size of $\hat{\boldsymbol{a}}_{i-1}^{\top}$ and the size of $\frac{\partial L}{\partial \boldsymbol{a}_i}$ may be different in each layer, Kronecker product is necessary.

2015) proposed to approximate the FIM through the Kronecker product of two smaller matrices to improve computational efficiency. Even so, FIM embedding for STE with complex decomposition methods is still not suitable for large-scale tasks, and the computational amount is large compared to the standard STE.

## 3.2 WEAK CURVATURE METRIC EMBEDDING FOR STE

### 3.2.1 INTRODUCTION OF WEAK CURVATURE MANIFOLD

We can use the Euclidean coordinates to calculate the natural gradient on the manifold because of the local homeomorphism of a manifold (Wald, 2010). Note that the homeomorphic mapping $\phi_U$ satisfies $U \in \mathcal{M} \mapsto \phi_U(U) \in \mathcal{R}^n$. For any point $x \in U$, we can define $\phi_U(x)$ as the Euclidean coordinate absolutely.

The FIM embedding for STE poses a significant challenge for computing. Intuitively, the computing $F^{-1}(\hat{\boldsymbol{\theta}})$ attached to the natural gradient will take on massive computation, which can only be numerically estimated. The inversion is mostly unrealistic when deep neural networks are very redundant, with tens of thousands of neural connections.

Inspired by information geometry (Amari, 2016) and mirror descent (Bubeck, 2015), we propose a weak curvature manifold with the weak curvature metric, which implies that the gradient mismatch problem can be alleviated by embedding the weak curvature manifold into the STE, while keeping a low computational complexity. Geometrically, the Riemannian manifold is nearly flat, where the Riemannian metric is an approximation of the Euclidean metric. In practice, we develop a linearized Riemannian metric from the Euclidean metric $\delta_{ij}$, which is systematically defined in general relativity (Wald, 2010).

**Definition 1** *(Weak Curvature Metric) Let a linearized Riemannian metric deviates from the Euclidean metric. The deviated metric $\epsilon_{ij}(\hat{\boldsymbol{\theta}})$ is much smaller than $1$ in global Euclidean coordinate system of $\delta_{ij}$,*

$$G_{ij}(\hat{\boldsymbol{\theta}}) = \delta_{ij} + \epsilon_{ij}(\hat{\boldsymbol{\theta}}), \quad \text{where } \left|\epsilon_{ij}\right| \ll 1. \tag{11}$$

*It is an adequate definition of "weakness" in this context and ensures $G_{ij}(\hat{\boldsymbol{\theta}})$ to be a positive-definite metric.*

### 3.2.2 HYPERBOLIC DIVERGENCE

In practice, we can determine a unique geodesic through exponential map $\exp_{\theta_i}(\tau\boldsymbol{\theta})$ that maps $\tau\boldsymbol{\theta}$ back to $\mathcal{M}$ where $\tau$ is a small constant, by defining $\tau\boldsymbol{\theta} \in T_{\theta_i}\mathcal{M}$ as the tangent vector. Note that the definition of exponential map is developed by (Wald, 2010; Helgason, 2001).

**Definition 2** *(Exponential Map) Let $\mathcal{M}$ be a Riemannian manifold, for the tangent vector $\boldsymbol{v} \in T_x\mathcal{M}$ in a point $x \in \mathcal{M}$ where $T_x\mathcal{M}$ is the tangent space, there is a unique geodesic $\gamma_{\boldsymbol{v}}(t)$ locally that satisfies $\gamma_{\boldsymbol{v}}(0) = x$ and $\gamma'_{\boldsymbol{v}}(0) = \boldsymbol{v}$. The exponential map $\exp_x : T_x\mathcal{M} \mapsto \mathcal{M}$ corresponding to $\gamma_{\boldsymbol{v}}(t)$ is defined as $\exp_x(\boldsymbol{v}) = \gamma_{\boldsymbol{v}}(1)$. When constraining $\exp_x$ to a neighbourhood $U$, this mapping is one-to-one.*

Empirically, the exponential map $\exp_{\theta_i}(\tau\boldsymbol{\theta})$ is to map a tangent vector $\tau\boldsymbol{\theta}$ in the tangent bundle to the point where the arc length from point $\theta_i$ is equal to $|\tau\boldsymbol{\theta}|$ on geodesic with the initial condition $(\theta_i, \tau\boldsymbol{\theta})$. In order to make $\exp(\tau\boldsymbol{\theta})$ and $\exp(-\tau\boldsymbol{\theta})$ have the same effect in the training process, we symmetrize them and derive a convex function

$$\psi(\boldsymbol{\theta}) = \sum_i \frac{1}{\tau^2} \log\left[\frac{\exp(\tau\theta_i) + \exp(-\tau\theta_i)}{2}\right] = \sum_i \frac{1}{\tau^2} \log(\cosh(\tau\theta_i)). \tag{12}$$

Geometrically, the convex function with geometric structure is introduced into the Bregman divergence (see Appendix B.2) (Bregman, 1967), and we obtain a novel Hyperbolic divergence that satisfies the criteria of divergence (see Appendix B.1).

**Definition 3** *(Hyperbolic Divergence) For a convex function $\psi$ defined by Eq. 12, the Hyperbolic divergence between $\boldsymbol{\theta}'$ and $\boldsymbol{\theta}$ is*

$$D_H[\boldsymbol{\theta}' : \boldsymbol{\theta}] = \sum_i \left[ \frac{1}{\tau^2} \log \frac{\cosh(\tau\theta_i')}{\cosh(\tau\theta_i)} - \frac{1}{\tau}(\theta_i' - \theta_i)\tanh(\tau\theta_i) \right]. \tag{13}$$

### 3.2.3 The Gradient Flow in Weak Curvature Manifold

**Lemma 2** *Let $d\boldsymbol{\theta} \to 0$, the square of an infinitesimal distance defined by Hyperbolic divergence is*

$$ds^2 = 2D_H[\boldsymbol{\theta}' : \boldsymbol{\theta}] \approx d\boldsymbol{\theta}^\top \left[\delta_{ij} - \tanh(\tau\boldsymbol{\theta})\tanh(\tau\boldsymbol{\theta})^\top\right] d\boldsymbol{\theta}. \tag{14}$$

*Furthermore, the defined metric $\left[\delta_{ij} - \tanh(\tau\boldsymbol{\theta})\tanh(\tau\boldsymbol{\theta})^\top\right]$ can be consistent with the weak curvature metric.*

*Proof.* The proofs can be found in Appendix B.4. Empirically, we can give a small constant $\tau$ to satisfy the weak curvature metric based on Definition 1. Comparing Eq. 4 and Eq. 11, we can deduce the deviated metric $\epsilon_{ij} = -\left[\tanh(\tau\boldsymbol{\theta})\tanh(\tau\boldsymbol{\theta})^\top\right]_{ij}$. □

**Lemma 3** *In the Riemannian manifold defined by Hyperbolic divergence, a weak curvature metric embedding for Straight-Through Estimator is approximated by:*

$$\tilde{\nabla}_{\boldsymbol{\theta}} L \overset{\text{STE}}{\approx} \left[\boldsymbol{I} + \tanh(\tau\boldsymbol{\theta})\tanh(\tau\boldsymbol{\theta})^\top\right] \nabla_{\hat{\boldsymbol{\theta}}} L \circ \mathbb{I}_{|\boldsymbol{\theta}| \leq 1} \tag{15}$$

*Proof.* The proofs can be found in Appendix C.2. □

## 3.3 Hierarchical Embeddings of the Weak Curvature Metric

When embedding a weak curvature manifold, there is an unnatural gap between the calculation of weak curvature metric and the layer-by-layer gradient update in back-propagation. Specifically, the metric is not a block diagonal matrix (each block corresponds to a layer of the neural network).

For the process of the quantization, we have the hierarchical embeddings for STE through decoupling operation:

$$\nabla_{\text{vec}(\boldsymbol{W}_i)} L \overset{\text{STE}}{\approx} \left[\boldsymbol{I} + \tau_i^2 \cdot \text{vec}(\tanh(\boldsymbol{W}_i))\text{vec}(\tanh(\boldsymbol{W}_i))^\top\right] \nabla_{\text{vec}(\hat{\boldsymbol{W}}_i)} L \circ \mathbb{I}_{|\text{vec}(\boldsymbol{W}_i)| \leq 1}$$

$$= \left(\boldsymbol{I} + \chi \frac{\text{vec}(\tanh(\boldsymbol{W}_i)\text{vec}(\tanh(\boldsymbol{W}_i))^\top}{\text{tr}\left[\text{vec}(\tanh(\boldsymbol{W}_i))\text{vec}(\tanh(\boldsymbol{W}_i))^\top\right]}\right) \nabla_{\text{vec}(\hat{\boldsymbol{W}}_i)} L \circ \mathbb{I}_{|\text{vec}(\boldsymbol{W}_i)| \leq 1}$$

$$= \begin{bmatrix} 1 + \chi\frac{\tanh^2(w_{i1})}{\text{tr}[\boldsymbol{P}_i]} & \chi\frac{\tanh(w_{i1})\tanh(w_{i2})}{\text{tr}[\boldsymbol{P}_i]} & \cdots \\ \chi\frac{\tanh(w_{i2})\tanh(w_{i1})}{\text{tr}[\boldsymbol{P}_i]} & 1 + \chi\frac{\tanh^2(w_{i2})}{\text{tr}[\boldsymbol{P}_i]} & \cdots \\ \vdots & \vdots & \ddots \end{bmatrix} \nabla_{\text{vec}(\hat{\boldsymbol{W}}_i)} L \circ \mathbb{I}_{|\text{vec}(\boldsymbol{W}_i)| \leq 1}.$$

$$\tag{16}$$

In the first line of the above formula, we can put $\tau_i^2$ outside because the elements of the deviated metric is very small based on Lemma 2. In the second line, we use a normalized formula to uniformly represent $\tau_i^2$: $\tau_i^2 = \chi/\text{tr}\left[\text{vec}(\tanh(\boldsymbol{W}_i))\text{vec}(\tanh(\boldsymbol{W}_i))^\top\right]$ where $\chi$ is a constant factor and the matrix $\left[\text{vec}(\tanh(\boldsymbol{W}_i))\text{vec}(\tanh(\boldsymbol{W}_i))^\top\right]$ is re-expressed by $\boldsymbol{P}_i$. By limiting the value range of full-precision weights to $[-1, +1]$, we obtain the independent weak curvature metric embedding in each layer. The algorithm of this weak curvature gradient with STE is illustrated in Appendix E.

## 4 Experiment

In this section, we implement experiments to demonstrate the effectiveness of our proposed methods on benchmark datasets that are CIFAR10/100 and ImageNet mainly here. Intuitively, experiments on CIFAR10/100 and ImageNet are ablation studies to validate the advantages of weak curvature manifold embeddings. Comparisons with other training-based quantization on quantitative indicates will be carried out on CIFAR10 and ImageNet. All experiments are conducted with PyTorch.

Table 1: The experimental results on CIFAR10 with ResNet20/32/44.

| Network | Forward | Backward | Test Acc (%) | Original Acc (%) |
|---|---|---|---|---|
| ResNet20 | $\{-1,+1\}$ | Dorefa
MultiFCG
WCG | 88.28±0.81
88.94±0.46
**89.78±0.33** | 91.50 |
| ResNet32 | $\{-1,+1\}$ | Dorefa
MultiFCG
WCG | 90.23±0.63
89.63±0.38
**90.52±0.27** | 92.13 |
| ResNet44 | $\{-1,+1\}$ | Dorefa
MultiFCG
WCG | 90.71±0.58
90.54±0.21
**91.38±0.11** | 93.56 |

## 4.1 EXPERIMENT SETUP

We use a weight decay of 1e-4, a batch size of 128, and SGD optimization method with nesterov momentum of 0.9. For CIFAR, we set total training epochs as 200 where the learning strategy is lowered by 10 times at epoch 80, 150, and 190, with the initial 0.1. For ImageNet, we set total training epochs as 50 where the learning strategy is lowered by 10 times at epoch 15, 30, and 45, with the initial 0.1. Note that we set $\chi$ as the reciprocal of output channel in a layer. All experiments are conducted for 5 times, and the statistics of the last 10/5 epochs' test accuracy are reported as a fair comparison.

Table 2: The experimental results on CIFAR100 with ResNet56/110.

| Network | Forward | Backward | Test Acc (%) | Original Acc (%) |
|---|---|---|---|---|
| ResNet56 | $\{-1,+1\}$ | Dorefa
MultiFCG
FCGrad
WCG | 66.71±2.32
66.58±0.37
66.56±0.35
**68.85±0.41** | 71.22 |
| ResNet110 | $\{-1,+1\}$ | Dorefa
MultiFCG
FCGrad
WCG | 68.15±0.50
68.27±0.14
68.74±0.36
**69.10±0.26** | 72.54 |

## 4.2 ABLATION STUDIES

In order to illustrate the superiority of the weak curvature gradient (WCG), we train QNNs with our gradient from scratch compared to other gradients, i.e. Dorefa (Zhou et al., 2016), MultiFCG (Chen et al., 2019) and FCGrad (Chen et al., 2019). We choose **1-bit** weight quantization for **all** layers in the networks, which means that each layer contains just $-1$ or $+1$. Note that the initialization is Xavier (Glorot & Bengio, 2010). Specifically, we experiment ResNet20/32/44 models (He et al., 2016) on CIFAR10, ResNet56/110 models (He et al., 2016) on CIFAR100 and ResNet18 model (He et al., 2016) on ImageNet. The accuracy of full-precision baseline is reported by (Chen et al., 2019).

The results of ablation studies on CIFAR10, CIFAR100 and ImageNet are shown in Table 1, Table 2 and Table 3 respectively. Various quantized models applying weak curvature gradients show significant improvement in challenging classification tasks. For fairness, we use the same forward quantization process and optimizations in experiments where the only difference is the gradients of the backward process.

Table 3: The experimental results on ImageNet with ResNet18.

| Network | Forward | Backward | Test Top1/Top5 (%) | Original Top1/Top5 (%) |
|---------|---------|----------|---------------------|------------------------|
| ResNet18 | $\{-1,+1\}$ | Dorefa
MultiFCG
FCGrad
WCG | 58.34±2.07/81.47±1.56
59.47±0.02/82.41±0.01
59.83±0.36/82.67±0.23
**61.02±0.33/83.74±0.18** | 69.76/89.08 |

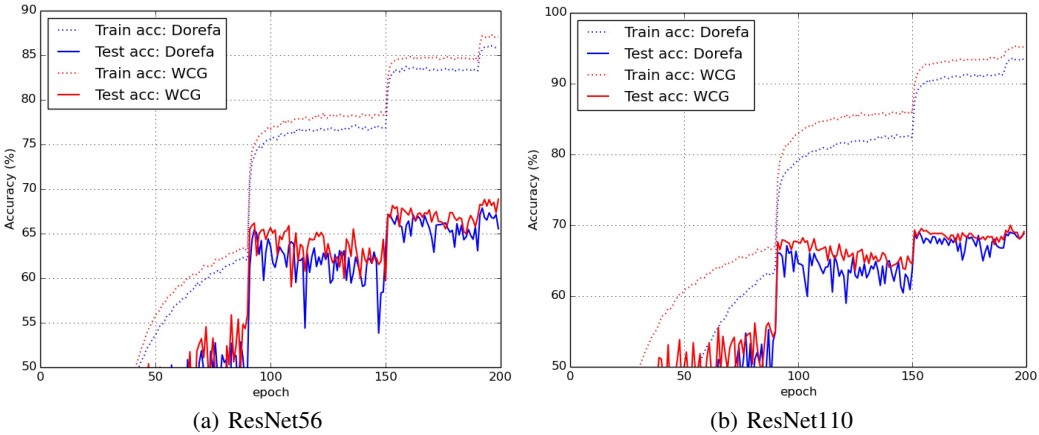

(a) ResNet56          (b) ResNet110

Figure 2: Training and test curves of ResNet56/110 on CIFAR100 compared between Dorefa and WCG.

### 4.3 Convergence and Stability Analysis

In this experiment, we compare the convergence and stability of Dorefa and WCG during the training process. As Figure 2 shows, the quantized model trained with WCG has better performance and more stable convergence than the quantized model trained with Dorefa, both for training and testing. Here, we state that better performance means larger mean, and more stable convergence means smaller variance. As for why WCG performs better, we conjecture that WCG has increased the Fisher information of the original quantized model compared to other methods, although WCG does not use additional neural network information for learning like MultiFCG or FCGrad. Moreover, this is where the advantage of manifold learning lies.

### 4.4 Comparisons with Other Training-based Quantization

In this experiment, our ManiQuant is initialized with the trained full-precision model, and continues to use WCG to learn the quantized model. ManiQuant focuses on the weak curvature manifold embeddings for STE, without any extra training tricks and the deformation of the quantization. We list the comparison results of 1-bit weight and activation quantization in Table 4. More experimental results can be found in Appendix D.

### 4.5 Training Time Analysis

Our hardware environment is conducted with an Intel(R) Xeon(R) Silver 4214 CPU(2.20GHz), GeForce GTX 1080Ti GPU. We test the training time per iteration as for MetaQuant using MultiFCG, ManiQuant using WCG, and Dorefa with ResNet20 in CIFAR10. For finishing one iteration of training, ManiQuant costs **18.44s** and MetaQuant costs **25.34s** while Dorefa uses **17.81s**.

Table 4: The experimental results on CIFAR10 with ResNet18 and VGG16.

| Network | W/A | Method | Test Acc (%) | Original Acc (%) |
|---------|-----|--------|--------------|-------------------|
| ResNet18 | 1/1 | RBNN (Lin et al., 2020) | 92.20 | 94.84 |
| | | MD (Ajanthan et al., 2021) | 91.28 | |
| | | PGD (Ajanthan et al., 2019) | 92.60 | |
| | | ManiQuant (WCG) | **92.83** | |
| VGG16 | 1/1 | RBNN (Lin et al., 2020) | 91.30 | 93.33 |
| | | MD (Ajanthan et al., 2021) | 90.47 | |
| | | PGD (Ajanthan et al., 2019) | 88.48 | |
| | | ManiQuant (WCG) | **91.77** | |

## 5 CONCLUSION

Empirically, training-based quantization methods tend to use Straight-Through Estimator to penetrate the non-differentiable quantization. However, we throw out the gradient mismatch problem in training quantized neural networks. In this paper, we introduce manifold learning to assist Straight-Through Estimator penetrating the non-differentiable quantization, which can be considered to alleviate the gradient mismatch problem. Specifically, we embed the Riemannian manifold into the Straight-Through Estimator to increase Fisher information. On the one hand, we try to embed Fisher Information Matrix for Straight-Through Estimator as the natural gradient defined by KL divergence, but encounter a complicated calculation dilemma. On the other hand, we propose a weak curvature manifold defined by Hyperbolic divergence to jump out the complicated calculation dilemma and embed weak curvature metric for Straight-Through Estimator as the weak curvature gradient.

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

# A STRAIGHT-THROUGH ESTIMATOR IN 1-BIT QUANTIZATION

The key of the proof is to divide the loss function into two parts based on conditional probabilities $P(a > \epsilon|a)$ and $1 - P(a > \epsilon|a)$:

$$
\begin{aligned}
\mathbb{E}_\epsilon \left[ \frac{\partial}{\partial a} L \right] &= \frac{\partial}{\partial a} \mathbb{E}_\epsilon[L] \\
&= \frac{\partial}{\partial a} \left[ L(\hat{a} = 1)P(a > \epsilon|a) + L(\hat{a} = -1)(1 - P(a > \epsilon|a)) \right] \\
&= \frac{\partial}{\partial a} P(a > \epsilon|a)[L(\hat{a} = 1) - L(\hat{a} = -1)],
\end{aligned}
\tag{17}
$$

where we divide the conditional probability $P(a > \epsilon|a)$ into two parts $|a| \leq 1$ and $|a| > 1$ based on the distribution of the noise source:

$$
\begin{aligned}
\frac{\partial}{\partial a} P(a > \epsilon|a) &= \frac{\partial}{\partial a} \left( P(a > \epsilon|a)\big|_{|a|>1} + P(a > \epsilon|a)\big|_{|a|\leq 1} \right) \\
&= \frac{\partial}{\partial a} \left( \int_{-1}^{1} \frac{1}{2} d\epsilon \bigg|_{|a|>1} + \int_{-a}^{a} \frac{1}{2} d\epsilon \bigg|_{|a|\leq 1} \right) = \mathbb{I}_{|a|\leq 1}.
\end{aligned}
\tag{18}
$$

Further, ignoring high-order derivative of a single neuron ($\hat{a} = \pm 1$) that typically has only a small impact on the total loss function, the loss function can be approximated by Taylor expansion w.r.t. $\hat{a} = 0$

$$
\begin{aligned}
L(\hat{a} = +1) &= L(\hat{a} = 0) + \frac{\partial L}{\partial \hat{a}}\bigg|_{\hat{a}=0} + \frac{1}{2} \frac{\partial^2 L}{\partial \hat{a}^2}\bigg|_{\hat{a}=0} + O\left( \frac{\partial^3 L}{\partial \hat{a}^3}\bigg|_{\hat{a}=0} \right), \\
L(\hat{a} = -1) &= L(\hat{a} = 0) - \frac{\partial L}{\partial \hat{a}}\bigg|_{\hat{a}=0} + \frac{1}{2} \frac{\partial^2 L}{\partial \hat{a}^2}\bigg|_{\hat{a}=0} + O\left( \frac{\partial^3 L}{\partial \hat{a}^3}\bigg|_{\hat{a}=0} \right).
\end{aligned}
\tag{19}
$$

Eq. 17 can be simply expressed as

$$
\mathbb{E}_\epsilon \left[ \frac{\partial L}{\partial a} \right] = \left( 2 \frac{\partial L}{\partial \hat{a}}\bigg|_{\hat{a}=0} \right) \mathbb{I}_{|a|\leq 1} = \frac{\partial L}{\partial \hat{a}} \mathbb{I}_{|a|\leq 1}.
$$

# B DIVERGENCE

## B.1 DEFINITION OF DIVERGENCE IN A MANIFOLD

$D[P : Q]$ is called a divergence when it satisfies the following criteria:

1) $D[P : Q] \geq 0$. 2) $D[P : Q] = 0$ when and only when $P = Q$. 3) When $P$ and $Q$ are sufficiently close, by denoting their coordinates by $\boldsymbol{\xi}_P$ and $\boldsymbol{\xi}_Q = \boldsymbol{\xi}_P + d\boldsymbol{\xi}$, the Taylor expansion of $D$ is written as

$$
D[\boldsymbol{\xi}_P : \boldsymbol{\xi}_P + d\boldsymbol{\xi}] = \frac{1}{2} \sum_{i,j} G_{ij}(\boldsymbol{\xi}_P) d\xi_i d\xi_j + O(|d\boldsymbol{\xi}|^3),
\tag{20}
$$

and Riemannian metric $G_{ij}$ is positive-definite, depending on $\boldsymbol{\xi}_P$.

## B.2 BREGMAN DIVERGENCE

Bregman divergence $D_B[\boldsymbol{\xi} : \boldsymbol{\xi}']$ is defined as the difference between a convex function $\psi(\boldsymbol{\xi})$ and its tangent hyperplane $z = \psi(\boldsymbol{\xi}') + (\boldsymbol{\xi} - \boldsymbol{\xi}')\nabla\psi(\boldsymbol{\xi}')$, depending on the Taylor expansion at the point $\boldsymbol{\xi}'$:

$$
D_B[\boldsymbol{\xi} : \boldsymbol{\xi}'] = \psi(\boldsymbol{\xi}) - \psi(\boldsymbol{\xi}') - (\boldsymbol{\xi} - \boldsymbol{\xi}')\nabla\psi(\boldsymbol{\xi}').
\tag{21}
$$

## B.3 KL DIVERGENCE AND FISHER INFORMATION MATRIX

The KL divergence can be defined between $p(\boldsymbol{x}|\boldsymbol{\theta})$ and $p(\boldsymbol{x}|\boldsymbol{\theta}')$:

$$
D_{KL}[p(\boldsymbol{x}|\boldsymbol{\theta}) : p(\boldsymbol{x}|\boldsymbol{\theta}')] = \int p(\boldsymbol{x}|\boldsymbol{\theta}) \log p(\boldsymbol{x}|\boldsymbol{\theta})d\boldsymbol{x} - \int p(\boldsymbol{x}|\boldsymbol{\theta}) \log p(\boldsymbol{x}|\boldsymbol{\theta}')d\boldsymbol{x}.
\tag{22}
$$

The first derivative is:

$$
\nabla_{\boldsymbol{\theta}'} D_{KL}[p(\boldsymbol{x}|\boldsymbol{\theta}) : p(\boldsymbol{x}|\boldsymbol{\theta}')]
$$
$$
= \int p(\boldsymbol{x}|\boldsymbol{\theta})\nabla_{\boldsymbol{\theta}'} \log p(\boldsymbol{x}|\boldsymbol{\theta})d\boldsymbol{x} - \int p(\boldsymbol{x}|\boldsymbol{\theta})\nabla_{\boldsymbol{\theta}'} \log p(\boldsymbol{x}|\boldsymbol{\theta}')d\boldsymbol{x} \tag{23}
$$
$$
= - \int p(\boldsymbol{x}|\boldsymbol{\theta})\nabla_{\boldsymbol{\theta}'} \log p(\boldsymbol{x}|\boldsymbol{\theta}')d\boldsymbol{x}.
$$

The second derivative is:

$$
\nabla_{\boldsymbol{\theta}'}^2 D_{KL}[p(\boldsymbol{x}|\boldsymbol{\theta}) : p(\boldsymbol{x}|\boldsymbol{\theta}')]
$$
$$
= \int p(\boldsymbol{x}|\boldsymbol{\theta})\nabla_{\boldsymbol{\theta}'}^2 \log p(\boldsymbol{x}|\boldsymbol{\theta})d\boldsymbol{x} - \int p(\boldsymbol{x}|\boldsymbol{\theta})\nabla_{\boldsymbol{\theta}'}^2 \log p(\boldsymbol{x}|\boldsymbol{\theta}')d\boldsymbol{x} \tag{24}
$$
$$
= - \int p(\boldsymbol{x}|\boldsymbol{\theta})\nabla_{\boldsymbol{\theta}'}^2 \log p(\boldsymbol{x}|\boldsymbol{\theta}')d\boldsymbol{x}.
$$

We deduce the Taylor expansion of the KL divergence at $\boldsymbol{\theta} = \boldsymbol{\theta}'$:

$$
D_{KL}[p(\boldsymbol{x}|\boldsymbol{\theta}) : p(\boldsymbol{x}|\boldsymbol{\theta}')] \approx D_{KL}[p(\boldsymbol{x}|\boldsymbol{\theta}) : p(\boldsymbol{x}|\boldsymbol{\theta})]
$$
$$
- \left( \int p(\boldsymbol{x}|\boldsymbol{\theta})\nabla_{\boldsymbol{\theta}} \log p(\boldsymbol{x}|\boldsymbol{\theta})|_{\boldsymbol{\theta}=\boldsymbol{\theta}'}d\boldsymbol{x} \right)^{\top} d\boldsymbol{\theta} - \frac{1}{2}d\boldsymbol{\theta}^{\top} \left( \int p(\boldsymbol{x}|\boldsymbol{\theta})\nabla_{\boldsymbol{\theta}}^2 \log p(\boldsymbol{x}|\boldsymbol{\theta})|_{\boldsymbol{\theta}=\boldsymbol{\theta}'}d\boldsymbol{x} \right) d\boldsymbol{\theta}
$$
$$
= 0 - \left( \int p(\boldsymbol{x}|\boldsymbol{\theta})\frac{\nabla p(\boldsymbol{x}|\boldsymbol{\theta})}{p(\boldsymbol{x}|\boldsymbol{\theta})}d\boldsymbol{x} \right)^{\top} d\boldsymbol{\theta} - \frac{1}{2}d\boldsymbol{\theta}^{\top} \left( \int p(\boldsymbol{x}|\boldsymbol{\theta})\nabla \left[ \frac{\nabla p(\boldsymbol{x}|\boldsymbol{\theta})}{p(\boldsymbol{x}|\boldsymbol{\theta})} \right] d\boldsymbol{x} \right) d\boldsymbol{\theta}
$$
$$
= - \left( \nabla \int p(\boldsymbol{x}|\boldsymbol{\theta})d\boldsymbol{x} \right)^{\top} d\boldsymbol{\theta} - \frac{1}{2}d\boldsymbol{\theta}^{\top} \left( \int p(\boldsymbol{x}|\boldsymbol{\theta})\frac{\nabla^2 p(\boldsymbol{x}|\boldsymbol{\theta})p(\boldsymbol{x}|\boldsymbol{\theta}) - \nabla p(\boldsymbol{x}|\boldsymbol{\theta})\nabla p(\boldsymbol{x}|\boldsymbol{\theta})^{\top}}{p(\boldsymbol{x}|\boldsymbol{\theta})p(\boldsymbol{x}|\boldsymbol{\theta})}d\boldsymbol{x} \right) d\boldsymbol{\theta}
$$
$$
= -(\nabla 1)d\boldsymbol{\theta} - \frac{1}{2}d\boldsymbol{\theta}^{\top} \left( \nabla^2 \int p(\boldsymbol{x}|\boldsymbol{\theta})d\boldsymbol{x} - \int p(\boldsymbol{x}|\boldsymbol{\theta}) \left[ \frac{\nabla p(\boldsymbol{x}|\boldsymbol{\theta})}{p(\boldsymbol{x}|\boldsymbol{\theta})} \right] \left[ \frac{\nabla p(\boldsymbol{x}|\boldsymbol{\theta})}{p(\boldsymbol{x}|\boldsymbol{\theta})} \right]^{\top} d\boldsymbol{x} \right) d\boldsymbol{\theta}
$$
$$
= \frac{1}{2}d\boldsymbol{\theta}^{\top}\mathbb{E}_{p(\boldsymbol{x}|\boldsymbol{\theta})}[\nabla \log p(\boldsymbol{x}|\boldsymbol{\theta})\nabla \log p(\boldsymbol{x}|\boldsymbol{\theta})^{\top}]d\boldsymbol{\theta}
$$
$$
= \frac{1}{2}d\boldsymbol{\theta}^{\top}F(\boldsymbol{\theta})d\boldsymbol{\theta}.
$$
$$\tag{25}$$

The Taylor expansion of the KL divergence at $\boldsymbol{\theta} = \boldsymbol{\theta}'$ is related to the Fisher Information Matrix.

## B.4 HYPERBOLIC DIVERGENCE AND WEAK CURVATURE METRIC

The first derivative of the Hyperbolic divergence is:

$$
\nabla_{\boldsymbol{\theta}'} D_H[\boldsymbol{\theta}' : \boldsymbol{\theta}]
$$
$$
= \sum_i \left[ \nabla_{\boldsymbol{\theta}'}\frac{1}{\tau^2} \log \cosh(\tau\theta_i') - \nabla_{\boldsymbol{\theta}'}\frac{1}{\tau^2} \log \cosh(\tau\theta_i) - \frac{1}{\tau}\nabla_{\boldsymbol{\theta}'}(\theta_i' - \theta_i)\tanh(\tau\theta_i) \right] \tag{26}
$$
$$
= \sum_i \nabla_{\boldsymbol{\theta}'}\frac{1}{\tau^2} \log \cosh(\tau\theta_i') - \frac{1}{\tau}\tanh(\tau\boldsymbol{\theta}).
$$

The second derivative of the Hyperbolic divergence is:

$$
\nabla_{\boldsymbol{\theta}'}^2 D_H[\boldsymbol{\theta}' : \boldsymbol{\theta}] = \sum_i \nabla_{\boldsymbol{\theta}'}^2 \frac{1}{\tau^2} \log \cosh(\tau\theta_i'). \tag{27}
$$

We deduce the Taylor expansion of the Hyperbolic divergence at $\boldsymbol{\theta} = \boldsymbol{\theta}'$:

$$
\begin{aligned}
D_H[\boldsymbol{\theta}' : \boldsymbol{\theta}] &\approx D_H[\boldsymbol{\theta} : \boldsymbol{\theta}] + \left( \sum_i \nabla_{\boldsymbol{\theta}'} \frac{1}{\tau^2} \log \cosh(\tau\theta_i') - \frac{1}{\tau} \tanh(\tau\boldsymbol{\theta}) \right)^\top \Bigg|_{\boldsymbol{\theta}'=\boldsymbol{\theta}} d\boldsymbol{\theta} \\
&+ \frac{1}{2} d\boldsymbol{\theta}^\top \left( \sum_i \nabla^2_{\boldsymbol{\theta}'} \frac{1}{\tau^2} \log \cosh(\tau\theta_i') \right) \Bigg|_{\boldsymbol{\theta}'=\boldsymbol{\theta}} d\boldsymbol{\theta} \\
&= 0 + 0 + \frac{1}{2\tau^2} d\boldsymbol{\theta}^\top \nabla \left[ \frac{\nabla \cosh(\tau\boldsymbol{\theta})}{\cosh(\tau\boldsymbol{\theta})} \right] d\boldsymbol{\theta} \\
&= \frac{1}{2\tau^2} d\boldsymbol{\theta}^\top \frac{\nabla^2 \cosh(\tau\boldsymbol{\theta}) \cosh(\tau\boldsymbol{\theta}) - \nabla \cosh(\tau\boldsymbol{\theta}) \nabla \cosh(\tau\boldsymbol{\theta})^\top}{\cosh^2(\tau\boldsymbol{\theta})} d\boldsymbol{\theta} \\
&= \frac{1}{2\tau^2} d\boldsymbol{\theta}^\top \left( \frac{\nabla^2 \cosh(\tau\boldsymbol{\theta})}{\cosh(\tau\boldsymbol{\theta})} - \tau^2 \left[ \frac{\sinh(\tau\boldsymbol{\theta})}{\cosh(\tau\boldsymbol{\theta})} \right] \left[ \frac{\sinh(\tau\boldsymbol{\theta})}{\cosh(\tau\boldsymbol{\theta})} \right]^\top \right) d\boldsymbol{\theta} \\
&= \frac{1}{2} \sum_{i,j} \delta_{ij} - \left[ \tanh(\tau\boldsymbol{\theta}) \tanh(\tau\boldsymbol{\theta})^\top \right]_{ij} d\theta_i d\theta_j.
\end{aligned}
\tag{28}
$$

## C  THE STEEPEST DESCENT DIRECTION IN A RIEMANNIAN MANIFOLD

### C.1  PROOF OF LEMMA 1

We would to know in which direction minimizes the loss function with the constraint of the KL divergence, so that we do the minimization:

$$
d\boldsymbol{\theta}^* = \underset{d\boldsymbol{\theta} \text{ s.t. } D_{KL}[p(\boldsymbol{x},\boldsymbol{\theta}):p(\boldsymbol{x},\boldsymbol{\theta}+d\boldsymbol{\theta})]=c}{\arg\min} L(\boldsymbol{\theta} + d\boldsymbol{\theta})
$$

where $c$ is the constant. The loss function descends along the manifold with constant speed, regardless the curvature.

Now, we write the minimization in Lagrangian form. Combined with Appendix B.3, the KL divergence can be approximated by its second order Taylor expansion. Approximating $L(\boldsymbol{\theta} + d\boldsymbol{\theta})$ with it first order Taylor expansion, we get:

$$
\begin{aligned}
d\boldsymbol{\theta}^* &= \underset{d\boldsymbol{\theta}}{\arg\min} \, L(\boldsymbol{\theta} + d\boldsymbol{\theta}) + \lambda \left( D_{KL}[p(\boldsymbol{x}, \boldsymbol{\theta}) : p(\boldsymbol{x}, \boldsymbol{\theta} + d\boldsymbol{\theta})] - c \right) \\
&\approx \underset{d\boldsymbol{\theta}}{\arg\min} \, L(\boldsymbol{\theta}) + \nabla_{\boldsymbol{\theta}} L(\boldsymbol{\theta})^\top d\boldsymbol{\theta} + \frac{\lambda}{2} d\boldsymbol{\theta}^\top F_{ij} d\boldsymbol{\theta} - c\lambda.
\end{aligned}
$$

To solve this minimization, we set its derivative w.r.t. $d\boldsymbol{\theta}$ to zero:

$$
\begin{aligned}
0 &= \frac{\partial}{\partial d\boldsymbol{\theta}} L(\boldsymbol{\theta}) + \nabla_{\boldsymbol{\theta}} L(\boldsymbol{\theta})^\top d\boldsymbol{\theta} + \frac{\lambda}{2} d\boldsymbol{\theta}^\top F_{ij} d\boldsymbol{\theta} - c\lambda \\
&= \nabla_{\boldsymbol{\theta}} L(\boldsymbol{\theta}) + \frac{\lambda}{2} F_{ij} d\boldsymbol{\theta} \\
\lambda F d\boldsymbol{\theta} &= -2 \nabla_{\boldsymbol{\theta}} L(\boldsymbol{\theta}) \\
d\boldsymbol{\theta} &= -\frac{2}{\lambda} F^{-1} \nabla_{\boldsymbol{\theta}} L(\boldsymbol{\theta})
\end{aligned}
$$

where a constant factor $2/\lambda$ can be absorbed into learning rate. Up to now, we get the optimal descent direction, i.e., the opposite direction of gradient which takes into account the local curvature defined by $F^{-1}$.

### C.2  PROOF OF LEMMA 3

Now we can deduce the steepest descent direction while taking into account the weak curvature manifold defined by the Hyperbolic divergence. With the constraint of Hyperbolic divergence in a

constant $c$, we do the minimization of the loss function $L(\boldsymbol{\theta})$ in Lagrangian form:

$$
\begin{aligned}
d\boldsymbol{\theta}^* &= \underset{s.t.D_H[\boldsymbol{\theta}':\boldsymbol{\theta}]=c}{\arg\min} L(\boldsymbol{\theta} + d\boldsymbol{\theta}) \\
&= \underset{d\boldsymbol{\theta}}{\arg\min} \, L(\boldsymbol{\theta} + d\boldsymbol{\theta}) + \lambda \left( D_H[\boldsymbol{\theta}' : \boldsymbol{\theta}] - c \right) \\
&\approx \underset{d\boldsymbol{\theta}}{\arg\min} \, L(\boldsymbol{\theta}) + \nabla_{\boldsymbol{\theta}} L(\boldsymbol{\theta})^\top d\boldsymbol{\theta} + \frac{\lambda}{2} d\boldsymbol{\theta}^\top \left[ \delta_{ij} - \tanh(\tau\boldsymbol{\theta})\tanh(\tau\boldsymbol{\theta})^\top \right] d\boldsymbol{\theta} - c\lambda.
\end{aligned}
\tag{29}
$$

To solve the above minimization, we set its derivative with respect to $d\boldsymbol{\theta}$ to zero and obtain the opposite direction of gradients in weak curvature manifold:

$$
\begin{aligned}
0 &= \nabla_{\boldsymbol{\theta}} L(\boldsymbol{\theta}) + \frac{\lambda}{2} \left[ \delta_{ij} - \tanh(\tau\boldsymbol{\theta})\tanh(\tau\boldsymbol{\theta})^\top \right] d\boldsymbol{\theta} \\
&\to -\frac{\lambda}{2} \left[ \delta_{ij} - \tanh(\tau\boldsymbol{\theta})\tanh(\tau\boldsymbol{\theta})^\top \right] d\boldsymbol{\theta} = \nabla_{\boldsymbol{\theta}} L(\boldsymbol{\theta}) \\
&\to d\boldsymbol{\theta} \approx -\frac{2}{\lambda} \left[ \boldsymbol{I} + \tanh(\tau\boldsymbol{\theta})\tanh(\tau\boldsymbol{\theta})^\top \right] \nabla_{\boldsymbol{\theta}} L(\boldsymbol{\theta}).
\end{aligned}
\tag{30}
$$

Where a constant factor $2/\lambda$ can be absorbed into learning rate. The last line of the above formula is true because each element in $\left[ \tanh(\tau\boldsymbol{\theta})\tanh(\tau\boldsymbol{\theta})^\top \right]$ is much smaller than 1 to satisfy the weak curvature metric. Specifically, $\left[ \boldsymbol{I} - \tanh(\tau\boldsymbol{\theta})\tanh(\tau\boldsymbol{\theta})^\top \right]$ is a strictly diagonally-dominant matrix which satisfies that the inverse matrix must exist.

## D    COMPARISONS WITH OTHER TRAINING-BASED QUANTIZATION

We first list the comparison results of 1-bit weight quantization in Table 5.

Table 5: The experimental results of ManiQuant compared to other quantization.

| Dataset | Network | Method | Test Acc (%) | Original Acc (%) |
|---------|---------|--------|--------------|------------------|
| CIFAR10 | ResNet20 | ProxQuant (Bai et al., 2018) | 90.21 | 91.50 |
| | | MetaQuant (Chen et al., 2019) | 90.80 | |
| | | LQ-Nets (Zhang et al., 2018) | 90.10 | |
| | | ManiQuant (WCG) | **90.88** | |
| ImageNet | ResNet18 | LR Net (Shayer et al., 2018) | 59.90/82.30 | 69.76/89.08 |
| | | MetaQuant (Chen et al., 2019) | 63.44/84.77 | |
| | | ADMM (Leng et al., 2018) | 64.80/86.20 | |
| | | ManiQuant (WCG) | **65.18/86.26** | |

Next, we extend the bit width to more than 1-bit, and the quantization function of k-bit used is

$$
Q(\boldsymbol{W}) = \frac{1}{2^{k-1} - 1} \operatorname{round} \left( (2^{k-1} - 1)\boldsymbol{W} \right).
\tag{31}
$$

We quantize the weights of all layers, and activations of all layers except the first layer, i.e, the image itself. Note that the QNN is initialized by the trained full-precision model. The comparison results are listed in Table 6.

Table 6: The classification accuracy results on ImageNet and comparison with other training-based quantizations, with AlexNet (Krizhevsky et al., 2012), ResNet-50 and MobileNet (Howard et al., 2017). Note that the accuracy of full-precision baseline is reported by (Elhoushi et al., 2019).

| Method | W | A | Top-1 (%) | | Top-5 (%) | |
|---|---|---|---|---|---|---|
| | | | Accuracy | Gap | Accuracy | Gap |
| **AlexNet** (Original) | 32 | 32 | 56.52 | - | 79.07 | - |
| ManiQuant (WCG) | 6 | 32 | 56.39 | $-0.13$ | 78.78 | $-0.29$ |
| DeepShift-Q (Elhoushi et al., 2019) | 6 | 32 | 54.97 | $-1.55$ | 78.26 | $-0.81$ |
| **ResNet-50** (Original) | 32 | 32 | 76.13 | - | 92.86 | - |
| ManiQuant (WCG) | 8 | 8 | 76.10 | $-0.03$ | 92.88 | $+0.02$ |
| INT8 (Zhu et al., 2020) | 8 | 8 | 75.87 | $-0.26$ | - | - |
| **MobileNet** (Original) | 32 | 32 | 70.61 | - | 89.47 | - |
| ManiQuant (WCG) | 5 | 5 | 61.32 | $-9.29$ | 84.08 | $-5.39$ |
| SR+DR (Gysel et al., 2018) | 5 | 5 | 59.39 | $-11.22$ | 82.35 | $-7.12$ |
| RQ ST (Louizos et al., 2019) | 5 | 5 | 56.85 | $-13.76$ | 80.35 | $-9.12$ |
| ManiQuant (WCG) | 8 | 8 | 70.86 | $+0.25$ | 89.60 | $+0.13$ |
| RQ (Louizos et al., 2019) | 8 | 8 | 70.43 | $-0.18$ | 89.42 | $-0.05$ |

## E  THE ALGORITHM DESIGN

---

**Algorithm 1** An algorithm for computing the gradient of the loss function $L$ and training a quantized neural network. Note that we use the gradient measured by the Hyperbolic divergence here. $\mathrm{Norm}(*)$ specifies how to normalize the activations and $\mathrm{BackNorm}(*)$ specifies how to backpropagate through the normalization, which is consistent with (Courbariaux et al., 2016). Note that $\mathrm{mat}(*)$ is the reverse operator of $\mathrm{vec}(*)$, which turns the column vector back into a matrix.

---

**Input:** A minibatch of inputs and targets $(\boldsymbol{x} = \boldsymbol{a_0}, \boldsymbol{y})$, $\boldsymbol{\theta}$ mapped to $(\boldsymbol{W}_1, \boldsymbol{W}_2, \ldots, \boldsymbol{W}_l)$, $\hat{\boldsymbol{\theta}}$ mapped to $\left(\hat{\boldsymbol{W}}_1, \hat{\boldsymbol{W}}_2, \ldots, \hat{\boldsymbol{W}}_l\right)$, a nonlinear function $f$, a constant factor $\chi$ and a learning rate $\eta$.

**Output:** The updated discrete parameters $\hat{\boldsymbol{\theta}}$.

1: {Forward propagation}
2: **for** $i = 1; i \leq l; i + +$ **do**
3:     Compute $\hat{\boldsymbol{W}}_i = Q(\boldsymbol{W_i})$;
4:     Compute $\boldsymbol{s}_i = \hat{\boldsymbol{W}}_i \hat{\boldsymbol{a}}_{i-1}$;
5:     Compute $\bar{\boldsymbol{a}}_i = \mathrm{Norm}\left(f \odot \boldsymbol{s}_i\right)$;
6:     Compute $\hat{\boldsymbol{a}}_i = Q(\bar{\boldsymbol{a}}_i)$;
7: **end for**
8: {Loss derivative}
9: Compute $\nabla_{\boldsymbol{a}_l} L = \frac{\partial L(\boldsymbol{y}, \boldsymbol{x})}{\partial \boldsymbol{z}}\big|_{\boldsymbol{z} = \hat{\boldsymbol{a}}_l} \mathbb{I}_{|\bar{\boldsymbol{a}}_l| \leq 1}$;
10: {Backward propagation}
11: **for** $i = l; i \geq 1; i - -$ **do**
12:     Compute $\nabla_{\boldsymbol{s}_i} L = \mathrm{BackNorm}(\nabla_{\boldsymbol{a}_i} L \odot f'(\boldsymbol{s}_i))$;
13:     Compute $\boldsymbol{P}_i = \mathrm{vec}\left(\tanh\left(\boldsymbol{W}_i\right)\right) \mathrm{vec}\left(\tanh\left(\boldsymbol{W}_i\right)\right)^{\top}$
14:     Compute $\tilde{\nabla}_{\boldsymbol{W}_i} L = \mathrm{mat}\left(\left(\boldsymbol{I} + \chi \cdot \frac{\boldsymbol{P}_i}{\mathrm{tr}[\boldsymbol{P}_i]}\right) \mathrm{vec}\left(\nabla_{\boldsymbol{s}_i} L\, \hat{\boldsymbol{a}}_{i-1}^{\top}\right)\right)$;
15:     Compute $\nabla_{\boldsymbol{a}_{i-1}} L = \hat{\boldsymbol{W}}_i^{\top} \nabla_{\boldsymbol{s}_i} L$;
16: **end for**
17: {The parameters update}
18: **for** $i = 1; i \leq l; i + +$ **do**
19:     Update $\boldsymbol{W}_i = \boldsymbol{W}_i - \eta \tilde{\nabla}_{\boldsymbol{W}_i} L \, \mathbb{I}_{|\boldsymbol{W}_i| \leq 1}$;
20: **end for**

---

