# OpenReview forum: "Riemannian Manifold Embeddings for Straight-Through Estimator"
_ICLR.cc/2022/Conference — ICLR 2022 Submitted_

### Official Review · Reviewer_gTNq · 2021-10-20

**Correctness:** 3
**Technical Novelty And Significance:** 2
**Empirical Novelty And Significance:** 3
**Recommendation:** 6
**Confidence:** 4

**Main Review:**

Pros:
1. The motivation of this paper is straight-forward and interesting: Basically, it first pointed out that quantization error is introduced by variance of STE, which can be upper bounded by the FIM. Then it related FIM with STE and kl-divergence.

Questions:
1. Author mentioned that FIM should be increased to alleviate the gradient mismatch problem. But the natural gradient introduced in Eq.10 does not seems to contribute to this goal. How the increase of FIM is related to the training ojective of model quantization?
2. How does optimizer interact with the proposed gradient? In experiments, author used SGD with momentum (SGD-M). Does that mean the natural gradient is firstly attained by Eq.18, then SGD-M is applied to the gradient to achieve gradient used in parameters update ? How if Adam is applied in the experiments since Adamm affects much to the initial gradient.
3. Besides, is SGD-M still applied in ImageNet experiments? Accuracy in table 3 is too high for SGD-M in ResNet18.

**Summary Of The Paper:**

This paper proposed to improve the gradient of quantization operator in quantization-aware training (QAT). Basically, it pointed out that the Euclidean assumption in general gradient (used by Straight-Through Estimator) can not reflect the curvature in the loss surface. Instead, it revised the gradient (natural gradient) based on mainfold learning. Due to the huge cost of Fisher Information Matrix (FIM) used to achieve natural gradient, it proposed to approximate it by weak curvature embedding.

**Summary Of The Review:**

It is interesting to incorporate manifold learning in QAT, but more concerete analysis (experiments) in deep learning field should be conducted.

---

### Official Review · Reviewer_HJGf · 2021-10-27

**Correctness:** 3
**Technical Novelty And Significance:** 3
**Empirical Novelty And Significance:** 3
**Recommendation:** 6
**Confidence:** 3

**Main Review:**

I find the paper clearly written. As a non-expert, I think that the questions that the gradient mismatch problem that the author aims to address is significant and the experiment results convincingly demonstrate the superior performance of the proposed gradient estimation method.

My main suggestion for the authors is to provide more context for the benefits of training QNN and position their contribution in this context. For instance, is the goal of training QNN to gain training speed, reduce GPU memory, or reduce the size of the network for storage purposes? How much can the proposed method improve upon these metrics (other than just accuracy), compared to full-precision networks? Indeed, the proposed method makes use of the natural gradients, which, to my knowledge, are much more computationally expensive to evaluate than first-order optimization methods such as the plain SGD. Can the proposed method improve training speed or memory usage upon a full-precision network trained with vanilla SGD? It will be great to have a table (similar to Table 2 and Table 3) to list the acceleration or storage gain of different methods and compare them with vanilla training of full-precision networks. There is a small subsection (subsection 4.5) that discusses the training time. This subsection, in my opinion, can be expanded, for instance, to include training time required for a full-precision network using SGD.

Minor: In the first paragraph of Section 3 on page 3, the authors mentioned that the quantization function is a **one-to-one** mapping from full-precision values to quantized values. Why is the quantization function a one-on-one map? I supposed that a single quantized value may correspond to multiple pre-quantized, full-precision values.

**Summary Of The Paper:**

This paper proposed a new gradient estimation method for training quantized neural networks (QNN). It alleviates the gradient mismatch problem that occurred in previous quantization methods that use the Straight-Through Estimator.


**Summary Of The Review:**

As a non-expert, I find that the method proposed in the paper is novel and empirically well-tested. However, the proposed approach for training QNNs seems to be computationally expensive on its own, and I am not sure if the required computational budget defeats the purpose of training QNN -- it will be great if the authors can help me understand in this regard.

---

### Official Review · Reviewer_Goby · 2021-11-01

**Correctness:** 2
**Technical Novelty And Significance:** 2
**Empirical Novelty And Significance:** 2
**Recommendation:** 3
**Confidence:** 3

**Main Review:**

In Theorem 1 (more specifically in the corresponding proofs in Appendix A), it seems Eq (8) is only $\textbf{approximately}$ unbiased, because of the Taylor expansion in Eq (21) in Appendix A. It's not rigorous to state the estimator in Eq (8) is unbiased. At least, this should be discussed carefully.

Honestly, I am confused about the underlying logic in Section 3.1. Of course, the gradient mismatch problem is caused by using the STE. But why does the root of that problem ONLY come from the STE variance (or its Cramer-Rao Lower Bound)?

In Eq (10), from the second row to the third row, you cannot simply replace the log-likelihood with the loss function L in general. Please elaborate on it.

In Figure 2, it seems the demonstrated results are from a single run. Please add the error bar that is shown in the tables.

Theoretically, why using the weak curvature gradient could deliver better performance?

**Summary Of The Paper:**

The main motivation for this paper is the gradient mismatch problem, which emerges from using the Straight-Through Estimator (STE) in training quantized neural networks and leads to an unstable training process. To deal with that gradient mismatch problem, the authors introduce the Manifold Quantization (ManiQuant) that embeds Riemannian manifolds into the STE. Specifically, the ManiQuant associates the gradient mismatch problem with Fisher information, which can then be exploited to alleviate that problem. Considering the high cost when inverting the Fisher information, the authors present an alternative simpler method related to Hyperbolic divergence and weak curvature manifold.



**Summary Of The Review:**

The underlying logic is not clear (see the detailed comments above).
Some statements/derivations are not rigorous and thus are not convincing.

---

### Official Review · Reviewer_WCKm · 2021-11-03

**Correctness:** 2
**Technical Novelty And Significance:** 3
**Empirical Novelty And Significance:** 3
**Recommendation:** 3
**Confidence:** 3

**Main Review:**

Strengths
- Investigating principled gradient estimators for the discrete quantization is an important problem to study.
- The idea of utilizing manifold geometry makes sense.

Weaknesses
- the actual used manifold in Sec. 3.3 does not seem to use the Fisher information. Why is the weak curvature metric better than the Euclidean distance?
- It looks like that the proposed estimator in Sec. 3.3 only applies to 1-bit quantization. How does the proposed approach generalize to multi-bit quantization?
- Some convergence theorems are required to justify that learning with the proposed estimator can indeed converge to better models.
- Is the vanilla straight-through estimator compared in the experiments?

**Summary Of The Paper:**

This paper proposes an alternative to straight-through estimator by considering the geometry of the likelihood. The proposed method can be viewed as a natural gradient descent algorithm on the Riemannian manifold. Compared to STE, the proposed estimator seems to penalize the quantities that are far from the quantization boundary. Slightly better accuracy results are reported for training 1-bit weight and activation neural networks.

**Summary Of The Review:**

The idea of the paper makes sense, but the theoretical and experimental results cannot support the usefulness of the proposed method well.

---

### Comment · Area_Chair_efy3 · 2021-11-08
**Too many errors, not technically sound**

The paper is difficult to read language-wise (grammatical errors, unclear, disconnected) and has too many technical errors, preventing understanding of the paper and making its theoretical statements uninterpretable or false. In my opinion, authors should have prepared the submission more carefully by proofreading it themselves or with the help of experienced colleagues, before submitting it for the review by external experts. For this reason I was proposing to PCs to desk-reject this submission, however there is no such practice at ICLR currently. The authors have an opportunity to receive full feedback from the assigned reviewers and discuss. Notwithstanding, at least a major revision and rewrite is necessary in order to meet clarity and correctness requirements. Despite there is a possibility to revise submissions within the review process, it would not be possible to reconsider such a major revision.

## Details
I will comment on the technical problems only, leaving the language issues aside (which is however still a major issue).
All equations should be numbered.

* The unnumbered equation on page 1: the Jacobian $d \hat W/d W$ is not found in the LHS therefore "where" is inappropriate.

* Second unnumbered equation on page 2: the chain rule is incorrect because the denominator in the first unnumbered equation on page 2 is not correctly differentiated.

* The presentation of the steepest descent and the natural gradient would be known to experts and can be shortened. In (5),(6) $x$ is not defined. The equation is given for a generative model and does not apply directly to supervised training.

* Page 3: Q is not one-to-one

### Theorem 1:

* $d L /d a$ is not defined.
* The first equation (19) does not hold because L is not continuously differentiable in a.
* It is not defined what the noise epsilon has to do with the neural network. In the current setting, L is independent of varepsilon and it must be $E_\varepsilon[L] = L$ and not the expansion in the second line of (19).
* In equation (20), the second term is expanded incorrectly, it should be integral from -1 to a. The result of (20) should be $1/2 I_{|a|\leq 0}$.
* The Taylor expansion at $\hat a = 0$ inappropriately leads to the gradient of the loss at  $\hat a = 0$ appearing in the unnumbered equation after (21). This is not the derivative at the quantized activation a as announced.
* Ignoring higher order terms of the Taylor expansion is not an accurate approximation because the Taylor expansion is used to approximate L(1)-L(-1), i.e. where the difference between arguments is large.

If this kind of derivation is done correctly, one obtains STE method for stochastic binary networks [r1,r2,r3], which is known to be biased unless the loss function is multilinear [r1].

### Definition 1:
* The gradient $d L /d \hat a$ is well defined and we need not estimate it. Not with $d L /d a$ for sure.
* The variable $x$ is not defined.
* No joint density function p(x, d L /d \hat a ) is defined. Instead we have a deterministic relation between $x$ (presumably the network output) and the gradient evaluated at quantized activation $\hat a$. It is not shown to satisfy necessary regularity conditions.
* The Fisher information matrix in (9) is different from FIM needed for the natural gradient descent. It seems that this definition is a mockup, with no factual use.
* The variance of the STE in the unnumbered equation on page 1 is in fact very easy to compute: because this STE is deterministic it is just zero.

### Lemma 1:
It should not be a lemma. It says: apply the natural gradient method with the gradient estimate computed by STE. There is nothing to proof. The expansion of the inverse FIM in (11) is by definition, it is completely general and has no use in the paper.

Considering the main (novel?) technical step in the paper (sec. 3.3 – 3.4), in short. It is not clear from the paper whether and how the hyperbolic divergence is related to the desired KL divergence or how its local form is related to the FIM.  Instead the convex function (14) seems to be introduced arbitrarily (or it is not explained). It seems that the authors arrive at a variant of mirror descent optimization in the space of continuous parameters, for which they used locally approximated divergence (15). While this is a valid choice for mirror descent, there is no connection with the natural gradient descent or the Rao-Cramer bound or the STE gradients and their respective errors. Formal statements, e.g. Lemma 3, unnecessarily bundle together independent components: (17) can be equally written for any estimate of the derivative.


[r1] Shekhovtsov, A., Yanush, V. „Reintroducing Straight-Through Estimators as Principled Methods for Stochastic Binary Networks“

[r2] Tokui, S. and Sato, I. „Evaluating the Variance of Likelihood-Ratio Gradient Estimators“

[r3] Dai, B. et al. „Stochastic Generative Hashing“

---

### Decision · Program_Chairs · 2022-01-20

**Decision:**

Reject

**Comment:**

The paper seeks to improve straight-through estimators by combining them with the ideas for correcting the step direction to be closer to a natural gradient.

While some (modest) improvements are demonstrated experimentally, the paper critically lacks technical correctness and has quite some gaps when trying to derive the algorithm from the natural gradient and Rao-Cramer bound. See public comments by reviewers and AC. The algorithm ends up to be a mirror descent with a mirror map, which is cheap to compute but not particularly well motivated. Moreover application of mirror descent to the activations (unlike the weights) is not well justified. The paper is rather unclear and hard to read also language-wise. Please proofread _before_ submitting.